# One-to-one befriending for people with intellectual disability and symptoms of depression: protocol for a pilot randomised controlled trial

Afia Ali  ,[1] Emma Mckenzie,[2] Angela Hassiotis,[1] Stefan Priebe,[3] Brynmor Lloyd-Evans,[1] Rumana Omar,[4] Rebecca Jones,[1] Monica Panca,[5] Vincent Fernandez,[6] Sally Finning,[6] Shirley Moore,[7] Danielle O'Connor,[7] Christine Roe,[7] Michael King[1]

**Correspondence to**
Dr Afia Ali; afia.ali@ucl.ac.uk

## ABSTRACT

**Introduction** People with intellectual disability (ID) are more likely to experience loneliness and have smaller social networks, which increases vulnerability to depression. Befriending may reduce depressive symptoms in other populations, but randomised controlled trials (RCTs) have not been carried out in this population. This pilot study aims to assess the acceptability and feasibility of carrying out a full RCT of one-to-one befriending by volunteers for people with ID, compared with an active control group.

**Methods and analysis** The trial aims to recruit 40 participants with ID. Participants in the intervention arm will receive weekly visits from a volunteer over 6 months. Community befriending schemes will recruit, train, supervise volunteers and match them to individuals with ID. Both groups will receive a booklet about local activities and have access to usual care. Health and social outcomes will be measured at the end of the intervention and 6 months' follow-up. The following outcomes will be assessed: (1) recruitment and retention of individuals with ID and volunteers in the trial, (2) adverse events related to the intervention, (3) the acceptability of the intervention, (4) whether the intervention is delivered as intended, (5) changes in health and social outcomes and (6) the feasibility of carrying out a cost-effectiveness analysis in a full trial. Qualitative data from participants, volunteers, staff and carers will identify barriers and facilitators of a future full trial.

**Ethics and dissemination** The study has been approved by the London City and East Research Ethics Committee (reference 18/LO/2188). The findings will be presented at conferences and published in a peer-reviewed journal and in the National Institute of Health Research journals library. A public engagement seminar will be held at the end of the study aimed at key stakeholders.

**Trial registration number** ISRCTN63779614.

## INTRODUCTION

Intellectual disability (ID) is a life-long condition characterised by impaired cognitive and adaptive functioning arising before the age of

### Strengths and limitations of the study

- ► This is the first pilot randomised controlled trial (RCT) of one-to-one befriending in people with intellectual disability.
- ► This pilot study will examine the feasibility and acceptability of carrying out a future RCT of befriending compared with an active control group.
- ► Community befriending services will be responsible for all aspects of the delivery of intervention.
- ► Availability of resources within befriending services may influence the delivery of the intervention.
- ► The findings will inform whether a full trial is feasible and what modifications would need to be made to overcome potential barriers.

18 years.[1] The UK prevalence of ID is 1%–2%.[2] People with ID have the same or higher prevalence of depression than the general population but are more likely to experience chronic depression.[3 4] They have greater exposure to social disadvantage,[5] experience social exclusion because of stigma,[6] have markedly smaller social networks[7 8] and a higher prevalence of loneliness compared with the general population.[9] These factors have been associated with depressive symptoms in this group.[10–13]

### Conceptualisation of befriending

Befriending is 'a relationship between two or more individuals, initiated, supported and monitored by an agency. The relationship is non-judgmental, mutual, purposeful, and there is a commitment over time'.[14] Key attributes are that it is a one-to-one friendship-like relationship; it is an organised intervention; and there is a negotiation of power.[15] There is a wide variation in the concept and practice of befriending.[16] At one extreme,



befriending is very close to a friendship, characterised as being reciprocal and is delivered by lay volunteers, and at the other end, it is a professional and therapeutic relationship, focused on the befriendee attaining goals and aspirations (mentoring). Most types of befriending relationships lie midway on this spectrum. Schemes are usually run by voluntary organisations in the community and offer training, supervision and ongoing support to volunteers.[16]

Befriending aims to help people who are lonely, isolated and have limited opportunities for social participation by increasing social and emotional support and by enhancing social networks and community participation. The causal mechanisms of befriending are unclear, but social support is thought to be important. Social support may act as a buffer to stress, and it may mediate genetic and environmental vulnerabilities to depression through its effects on neurobiological factors and other psychosocial factors (eg, coping strategies).[17] The befriender may enhance social support and link the befriendee into social activities, which may be sustainable outside of and beyond the end of the befriending relationship, which may result in longer-term benefits. Befriending may also improve health outcomes through its effects on social networks.[18]

There are also potential benefits for befrienders. Motivation for befriending often includes a desire for 'giving' something back to the community (eg, helping others) and 'getting' something in return (eg, acquiring new skills or meeting new people).[19 20] Positive benefits reported by befrienders include increased self-esteem and confidence, feeling that they have gained a genuine friend themselves and improved attitudes towards people with mental illness.[19] Volunteering in general has beneficial effects on depression, psychological well-being and life satisfaction, and is associated with lower risk of mortality, although the causal mechanisms for these associations are unclear.[21]

However, the benefits of befriending may be short-lived, and people with ID have reported feeling distressed following the termination of their befriending relationship.[22] Other risks include the emotional turbulence that is associated with a natural friendship, or harmful effects if the befriender is not adequately trained or supervised. There may also be undue burden placed on the befriender to take on excessive responsibility.[16]

### Effectiveness of befriending
One meta-analysis found that befriending in people with mental or physical health problems (delivered by social and healthcare professionals, as well as lay volunteers) had a statistically significant but modest effect on reducing symptoms of depression when compared with no treatment or treatment as usual in both the short and long term.[23] Another systematic review examined a range of health and social outcomes in studies where befriending was delivered by volunteers only.[24] Befriending was associated with better patient-reported outcomes when all primary outcomes were combined, but the effect size was

small.[24] However, in contrast to the previous review, there was limited evidence for the effectiveness of befriending on individual outcomes, such as depression, loneliness or quality of life, when the studies were combined.[24] A recently published randomised controlled trial (RCT) of befriending by volunteers in people with psychosis provides further evidence for the potential beneficial effects of befriending.[25] Participants in the intervention arm had significantly more social contacts at the end of the 12-month intervention and at 6 months' follow-up, suggesting that befriending may help to reduce social isolation in this is group.

The effectiveness of befriending in people with ID has not been evaluated in a randomised trial. An unpublished single-arm feasibility study of one-to-one befriending by volunteers, conducted by a voluntary organisation,[26] recruited 24 volunteers, of which 15 were matched with an individual with ID. Positive change was reported in 60% of individuals with ID; 53% reported a decrease in isolation, and 40% reported an increase in confidence. One Australian study examined the feasibility of using 'active mentoring', whereby members of existing community groups were trained to act as mentors to older adults with ID in order to provide social support and to encourage participation in community groups.[27] The intervention comprised 29 individuals receiving the intervention and a matched comparison group. The participants in the intervention reported better social satisfaction compared with the comparison group, but there were no significant changes in the other outcomes.

Given the dearth of studies examining befriending in people with ID, there is a clear rationale for carrying out a pilot study prior to a full RCT.

### AIMS AND OBJECTIVES
The main aim of the study was to determine the feasibility and acceptability of a full-scale RCT of one-to-one befriending by volunteers for people with ID in addition to usual care, compared with an active control arm.

The objectives were to
► Examine the recruitment and retention of individuals with ID and volunteers in the trial and the number of successfully matched pairs within the 6-month study recruitment period.
► Record any negative consequences/adverse effects of befriending.
► Measure the extent to which the intervention is delivered as intended by volunteers and the befriending schemes.
► Examine the acceptability of the intervention and study procedures by exploring the views of individuals, volunteers, carers and befriending services.
► Examine changes in health and social outcomes by carrying out exploratory analyses of the impact of befriending on depressive symptoms measured by the Glasgow Depression Scale[28] at 12 months and other outcomes (psychological distress, self-esteem,

loneliness, quality of life and social participation) at 6 and 12 months postrandomisation.
► Carry out exploratory analyses of the impact of befriending on volunteers' well-being, self-esteem, loneliness and attitudes towards people with ID at 6 and 12 months.
► Estimate the sample size required and determine the final trial design for a full-scale RCT.
► Assess the feasibility of collecting data that would inform a future analysis of cost effectiveness.

## METHODS
### Design
This is a two-arm, parallel group, researcher blind pilot RCT with 1:1 allocation. Fifty participants with ID who are eligible for the study will be randomly allocated to either the intervention arm (one-to-one befriending by a volunteer) or an active control arm. Both groups will have access to usual care and a booklet of local resources. The duration of the intervention will be 6 months. Outcome assessments will be carried out at baseline, postintervention and at 6 months' follow-up. A process evaluation, based on mixed methods, will be carried out to examine the delivery and adherence to intervention, and stakeholder views on the acceptability of the intervention and barriers and facilitators that may affect the implementation of a full-scale trial.

### Sample size
We do not have any estimates of the number of people with ID who are eligible and are likely to consent to taking part in the trial. If we approach 50 participants who are eligible to take part, this will allow us to estimate an expected recruitment rate of 80% (40 people), with a 95% CI of 68.9% to 91.1%. A sample size of 40 recruited people with ID would allow us to estimate a 30% drop-out rate in the trial with a 95% CI of 25.7% to 54.3%. The recruitment period is 6 months. There are two participating befriending services and therefore we will need to recruit 3.3 participants with ID per month at each site. Twenty volunteers will need to be recruited and matched to each participant in the intervention arm.

### Participants
#### Inclusion criteria
We will include individuals with ID aged 18 years or over who have mild or moderate ID (IQ of 35–60) assessed using the Wechsler Abbreviated Scale of Intelligence, Second Edition,[29] should not be attending college/education or a day service for 3 or more days a week; will have a score of 5 or more on the Glasgow Depression Scale for People with Learning Disability (GDS-LD[28]), indicating the presence of depressive symptoms but do not need to have a clinical diagnosis of depression. Participants will need to be able to speak English and provide informed consent.

Volunteers will be aged 18 years or over, who can agree to being available once a week for at least 1 hour over a period of 6 months. Volunteers do not need to have any prior experience of supporting people with ID.

### Exclusion criteria
Individuals with ID will be excluded if they have limited communication and comprehension and therefore would not be able to complete the questionnaires.

Volunteers will be excluded if they have a criminal record (any documented offence due to the vulnerability of this population) recorded on their Disclosure and Barring Service (DBS) check or are unable to provide two references or have unsuitable references.

### Recruitment
#### Befriending services
Two befriending services ('Outward', based in Hackney, North London, and 'the befriending scheme' in Suffolk) will be taking part in the study and will be responsible for overseeing the befriending intervention. Both services have experience supporting befriending relationships with people with ID, including matching individuals with volunteers and monitoring relationships.

#### Participants with ID
Participants with ID will be recruited from existing and new referrals to the befriending services or recruited from the caseload of clinicians working in community learning disability services at the North East London NHS Foundation Trust (NELFT) and Suffolk. Volunteer coordinators and clinicians will screen referrals and caseloads for participants who may be eligible for the study and will make the initial approach. If the individual is interested in the study and consents to his/her details being passed on to the research team, a referral form will be completed and emailed to the research team. The research assistant (RA) will then contact the individual and carry out an eligibility assessment. If the individual is eligible, they will need to provide written consent before completing the baseline assessment.

#### Volunteers
The befriending services will advertise and recruit volunteers through newspaper advertisements, befriending and job websites, social media, and recruitment events at colleges and universities. Volunteers will be recruited directly from the study website hosted by the UCL Division of Psychiatry and through the Division's twitter account. Interested volunteers will complete an application form and will be invited to an informal interview to assess their suitability and motivation for taking part in the scheme. A DBS check will be completed to ensure that they have no criminal records, and references will be obtained. People with ID are potentially a vulnerable group, and therefore, volunteers with any previous offence, including minor offences, will not be included in the study. Successful candidates will be invited to take part in the study. They will receive an information sheet and will be asked to sign a consent form, followed by completion of the baseline assessment.

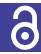

## Randomisation

Randomisation will be carried out after the baseline assessments have taken place. Participants will be randomised into the study by an unblinded member of the research team who will enter the patients' details into a web-based randomisation system, which is hosted on a secure server by sealed envelope. This system will randomly allocate the participant to either the intervention or control arm. The unblinded researcher will notify the befriending service and participants with ID of their allocation.

Randomisation will be blocked using randomly varying block sizes, stratified by centre. The allocation schedule will be concealed through the use of this central web-based randomisation service. The randomisation protocol will be created by the trial statistician and the set-up of the service will be overseen by the Priment Clinical Trials Unit.

## Intervention group

Participants with ID will meet with the volunteering coordinator to obtain information about their hobbies and interests and activities they would like support to participate in. Based on this information, participants with ID will be matched to a volunteer who can accommodate the person's interests.

### Befriending intervention

The befriending intervention has been adapted from the existing models used by the two participating befriending services and from other studies of befriending.[26 30] The purpose of the befriending relationship will be to provide friendship and emotional support, and to assist the individual to access activities in the community that they may be unable to do alone. As participants become more confident, they will be encouraged to access activities in the community on their own in order to promote sustained social activities beyond the befriending relationship.

The volunteer (befriender) and person with ID will meet once a week for at least 1 hour, for 6 months, although some breaks are anticipated due to holidays or illness. They will receive a booklet detailing local activities and amenities, which they can use to plan activities. The emphasis will be on assisting the individual to make choices about the activities that they wish to pursue. The volunteer is not expected to carry out personal care, administer medication or accompany the individual to medical appointments. Contacts by phone/social media can take place, in addition to face-to-face contacts. The pair can spend some sessions in the person's home, but this should not exceed 50% of the total number of sessions. Sessions may take place during evenings/weekends, depending on the pair's availability. They will keep a record of their activities in a structured log that will be provided (whether they attended each session, reasons for cancellation, what they did in each session and duration of activity) and record of other types of contact. Volunteers will be reimbursed travel expenses, but other expenses will need to be agreed with the befriending

service. Participants with ID will not be reimbursed travel expenses or the costs of activities.

### Introduction and monitoring of the befriending relationship

The volunteering coordinator will arrange a face-to-face meeting where the pair will be introduced to each other. If they agree to continue, the pair will arrange to meet on their own; if they decide that the pairing is unsuitable, they will be rematched. If the volunteer or individual with ID drops out of the relationship once it has become established, attempts will be made to rematch them. Volunteers may be matched to more than one participant with ID.

The volunteering coordinator will arrange a face-to-face meeting with the pair after 6 weeks and will maintain contact with each person by telephone/face-to-face contact every 4 weeks thereafter to monitor the progress of the relationship. A further meeting will be held with the pair at the end of the 6 months to obtain general feedback about the befriending intervention, to discuss ending the relationship and to support the individual with ID with coming to terms with the ending. The pair may continue their relationship if they wish after the 6-month period, but arrangements for monitoring the relationship will vary, depending on the befriending service. Information will be obtained on any relationships continuing beyond 6 months and the monitoring that has been provided.

### Training and supervision of volunteers

The volunteers will attend training delivered face-to-face and as e-learning. The training will cover the benefits of befriending and issues related to confidentiality and lone working; advice on how to plan meetings effectively; health and safety; safeguarding ; learning disability awareness; and professional boundaries, which covers dealing with sensitive issues, ending relationships and expectations of the role of the volunteer. Volunteers will also receive slides and a manual developed by the research team, with information about the study and additional information about communication, challenging behaviour and mental health problems.

Volunteers will have access to group or individual supervision delivered face-to-face or over the phone, once a month, by the volunteer coordinator, which will address issues that may have arisen from the relationship, for example, concerns about mental health or behaviour.

### Training and support for befriending services

Participating befriending services will complete good clinical practice (GCP) training and will undergo training on the study research processes and procedures as part of the site initiation visit. Volunteer coordinators will receive support from the research team through regular email and telephone contact, and they will attend Trial Management Group (TMG) meetings.

## Control group

Participants in the control arm will receive the activities booklet. They will meet with a member of the research team who will discuss the booklet with them (and their

carrer, if present) and encourage them to engage with activities. This is to control for the effects of participants in the treatment arm receiving more information about local activities.

Both the control and intervention arms will also have access to 'usual care'. This will include access to multidisciplinary input from community ID services. Participants can continue to take their usual medication and can access other community and hospital health services and day services. Information about usual care will be collected as part of the baseline assessment.

## Outcomes

### Recruitment and retention of participants

We will examine the proportion of participants with ID recruited from among those eligible, the proportion of volunteers recruited from among people expressing an interest over a 6-month period, the proportion of participants with ID who are successfully matched with a volunteer, the proportion of participants with ID and volunteers who drop out of the intervention arm and the proportion of participants and volunteers who complete subsequent follow-up assessments.

### Adverse events

Adverse events will be collected at each follow-up assessment using open-ended questions and will also be reported directly to the chief investigator by the befriending services, including concerns about safeguarding. Volunteers will follow a protocol if they have concerns about the

participant's mental health, which will involve informing the volunteer coordinator, who will in turn notify the CI and the referring clinician or a health professional involved in the person's care. If a serious incident (eg, self-harm and challenging behaviour) occurs out of hours, the carer will be notified and emergency services will be contacted if appropriate. All adverse events will be recorded in the medical records and Case Report Forms. Serious adverse events (eg, safeguarding concerns and hospitalisation) will be recorded in the serious adverse effects log.

### Acceptability of the intervention

This will be informed by data on retention/drop-out of volunteers and participants, the extent of engagement with the intervention by participants and volunteers (based on number of sessions attended) and qualitative data obtained from volunteers, participants with ID, carers of people with ID and staff from the befriending service.

### Adherence to the intervention

Data will be collected on volunteer training, uptake of supervision and the frequency of monitoring checks carried out by volunteer coordinators from routine records at each site in order to assess fidelity to the intervention by the befriending services. Structured logs provided by volunteers will be analysed to assess fidelity to the intervention by volunteers. We will examine (1) how many sessions were attended by each volunteer, reasons

| Table 1 Schedule of assessments in participants with ID | | | |
|---|---|---|---|
| Assessment/outcome | Screening and baseline assessment | Postintervention assessment | 6 months' follow-up |
| Eligibility confirmation WASI-II GDS-LD | x | | |
| Informed consent | x | | |
| Sociodemographic questionnaire | x | | |
| Adapted Rosenberg Self-Esteem Scale | x | x | x |
| MANS-LD | x | x | x |
| WHO-QOL-8 | x | x | x |
| MWLQ | x | x | x |
| SSSR | x | x | x |
| GCPLA | x | x | x |
| EQ-5D-Y | x | x | x |
| CSRI | x | x | x |
| Adverse effects review | x | x | x |
| Concomitant medication | x | x | x |
| Semistructured interview (optional) | | | x |

CSRI, Client Services Receipt Inventory; EQ-5D-Y, EuroQol-Youth; GCPLA, Guernsey Community Participation and Leisure Assessment; GDS-LD, Glasgow Depression Scale for People with Learning Disability; MANS-LD, Maslow Assessment of Needs Scale–Learning Disability; MWLQ, Modified Worker Loneliness Questionnaire; SSSR, Social Support Self-Report for Intellectually Disabled Adults; WASI-II, Wechsler Abbreviated Scale of Intelligence, Second Edition; WHO-QOL-8, WHO Quality of Life Questionnaire.

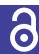

for non-attendance, and how many attended at least 10 sessions during the 6 month intervention period; and (2) for how many participants, the minimum threshold of at least 50% of meetings being outside the participant's home was achieved.

### Depression and other health and social outcome measures in participants with ID

All the health and social outcome measures have been validated in people with ID and will be assessed with the participant at baseline, at the end of the intervention and at 6 months' follow-up (see table 1 for schedule of assessments). Assessments will be carried out by an RA blind to group allocation and will take place face-to-face at the befriending service or the participants' homes.

► Depressive symptoms will be measured using the GDS-LD.[28] Scores at 6 months follow-up will be the main health outcome of interest.

► Self-esteem will be measured using the adapted Rosenberg Self-Esteem Scale for people with intellectual disabilities.[31]

► Quality of life will be measured using the Maslow Assessment of Needs Scale–Learning Disability[32] and five items from the adapted WHO Quality of Life Questionnaire.[33]

► Loneliness and social satisfaction will be measured using the Modified Worker Loneliness Questionnaire.[34]

► Social support will be measured using the Social Support Self-Report for Intellectually Disabled Adults.[35]

► Social participation will be measured using the Guernsey Community Participation and Leisure Assessment.[36]

### Volunteer outcome measures

The following outcome assessments will be carried out in volunteers at baseline (prior to matching) and postintervention (6 months after the baseline) and at 6 months' follow-up (see table 2 for schedule of assessments):

► Self-esteem will be measured using the 10-item Rosenberg Self-Esteem Scale.[37]

► Psychological well-being and quality of life will be measured using the Warwick-Edinburgh Mental Well-being Scale.[38]

► Loneliness will be measured using the UCLA Loneliness Scale.[39]

► Attitudes of volunteers will be assessed using the 67-item, Attitudes Towards Intellectual Disability Questionnaire.[40]

### Feasibility of carrying out a cost-effectiveness analysis

The preliminary health economic analysis will inform planning of future economic analyses, sources of data required and how best to collect these data. The following measures will be recorded at baseline, postintervention (6 months after randomisation) and at 6 months' follow-up.

► Health-related quality of life will be measured using the EuroQol-Youth (EQ-5D-Y).[41] We will assess the feasibility of using the EQ-5D-Y to calculate quality-adjusted life years (QALYs). QALYs will be calculated using the EuroQol- 5 Dimensions-3 levels (EQ-5D-3L) tariff, as there are no value sets for the EQ-5 D-Y.[42]

► Client Services Receipt Inventory (CSRI) for people with intellectual disabilities adapted for the study,[43] completed with carers where possible, will assess the feasibility of collecting participant-reported service use over the previous 6 months.

### Process evaluation

A process evaluation will be undertaken, based on MRC guidance.[44] The aim will be to examine whether the different components of the intervention were consistently followed; to examine the extent to which volunteers delivered the intervention as intended; to understand the perceived value, benefits and unintended consequences of the intervention so that these are fully measured in a full trial; to determine the contextual factors influencing how the intervention was delivered; and to develop an

| Table 2 Schedule of assessments with volunteers | | | |
|---|---|---|---|
| Assessment/outcome | Screening and baseline | Postintervention assessment | 6 months' follow-up |
| Eligibility check<br>  Informal interview<br>  DBS check references | x | | |
| Informed consent | x | | |
| Rosenberg Self-Esteem Scale | x | x | x |
| WEMWBS | x | x | x |
| UCLA Loneliness Scale | x | x | x |
| ATTID | x | x | x |
| Focus group (optional) | | | x |

ATTID, Attitudes Towards Intellectual Disability Questionnaire; WEMWBS, Warwick-Edinburgh Mental Well-being Scale.

understanding of the likely mechanisms of action of the intervention. We will interview 16 participants with ID after their 12-month follow-up (eight per site) and carry out two focus groups at each site with five to eight volunteers, staff from each befriending service and five to eight carers. We will use purposive sampling in order to include people with a range of demographic characteristics (eg, gender, age, ethnicity and living arrangements), and both participants and volunteers for whom the befriending relationship broke down, as well as people who completed the intervention. All respondents will be asked about what aspects of the intervention were perceived to be helpful, what aspects require modification and suggestions for improvement, views about trial procedures, and the perceived barriers and facilitators to delivering a larger trial. The interviews and focus groups will be audio-taped and transcribed.

In order to understand how the befriending intervention was delivered, the location, content of the meetings and the range of activities undertaken will be described based on an analysis of structured logbooks. A framework will be developed to categorise different types of activity to enable different types of befriending support to be distinguished and quantified. Test procedures will be developed for collecting quantitative process data which, in a future larger trial, could be used to explore the relationships between process variables and outcomes. Findings from the process evaluation will inform any necessary refinements to the study intervention or procedures.

## Analysis

Feasibility outcomes, such as the number of participants who were screened and eligible, the proportion of eligible participants with ID and interested volunteers recruited to the study, and the proportion successfully matched in befriending relationships, will be reported. The number (proportion) of drop-outs of volunteers and participants from the intervention arm and from both arms at each follow-up assessment will also be reported, including reasons why where possible. This information will be presented in a ConsolidatedStandards of Reporting Trials diagram describing the flow of participants through the study (http://www.consort-statement.org/).

The characteristics and outcomes of participants by trial arm will be summarised using means and SD or medians and IQRs for continuous variables and count and percentages for categorical variables. Appropriate regression models, depending on the type of outcome, adjusted for baseline values and centre will be used to estimate the intervention effect on health and social outcomes where possible. The results will be presented as estimates with 95% CI. All analyses will be carried out as randomised with available data (intention-to-treat principle) and will be used to inform the definitive trial. The characteristics of patients with missing outcome data will be investigated.

## Economic evaluation

We will assess the feasibility of gathering information for a cost-effectiveness analysis for a full RCT, including testing the suitability of calculating QALYs using the EQ-5D-Y. We will calculate the costs of delivering the befriending intervention (training, supervision and expenses), which will be obtained from each participating befriending service. Mean resource use/costs (SD) per participant at baseline and follow-up, based on completion of the CSRI, will be presented.

## Qualitative analysis

Transcripts from the focus groups and interviews will be analysed by the study team using thematic analysis supported by computer software (NVivo V.9). The analytical strategy will focus on identifying barriers and facilitators to implementing the intervention successfully, its benefits to participants and mechanisms of effect. Analysis will also allow consideration of themes that arise more inductively from the data. Validity will be enhanced by a collaborative analytical strategy involving members of the research team and the advisory group who will meet together to review the coding framework and agree on the themes.

## Patient and public involvement

Volunteers, people with ID and befriending schemes were involved in the design of the study. During the study, there will be advisory groups comprising carers, current volunteers and individuals with ID who will provide advice about the study information sheets, consent forms, topic guides for the qualitative interviews and focus groups, results of the study findings and the final study report. Members will be invited to be part of the TMG and will be involved in carrying out the qualitative interviews and focus groups as part of the process evaluation. They will undergo training and support for this role. We will also invite members to participate in the public engagement seminar at the end of the study.

## Ethics, governance and dissemination

Amendments to the study protocol and documents will be approved by the sponsor (UCL) and the ethics committee. Priment Clinical Trials Unit (CTU) will ensure that the trial procedures meet the requirements of GCP and will complete data quality assurance checks. The RA will have experience working with people with ID and will undergo training on assessing capacity and carrying out assessments. Supervision of the RA will be provided by the chief investigator (AA). The study team, coapplicants, Patient and Public Involvement (PPI) advisors and the CTU will be part of the TMG and will meet every 10–12 weeks to discuss the progress of the study. An independent trial steering committee will provide overall supervision of the trial and will report to the funders. Data confidentiality will be maintained by assigning participants study identification numbers, and data will be entered anonymously and stored in a secure web-based database.

The findings of the study will be presented at conferences and published in an open-access peer-reviewed journal, as well as the National Institute of Health Research Public



Health Research Journal. A public engagement seminar targeted at relevant stakeholders will take place at the end of the study to discuss the study findings and implications. A summary report will be developed for participants and volunteers taking part in the study and will be published on the study website.

### Study timeline

Recruitment of participants began in April 2019. There will be a 6-month recruitment period, a further 12 months to complete the follow-up assessment, and 3 months to complete the process evaluation, analyse the results and write up the study findings.

## DISCUSSION

This is the first pilot RCT of one-to-one befriending, monitored by community befriending services, in people with ID. The trial will provide data on whether a full trial will be feasible, in terms of recruitment and retention of volunteers and people with ID and data on potential beneficial and adverse effects, acceptability and the extent to which the intervention is delivered as intended by volunteers and the befriending services. It will help to inform modifications that need to be made to enable a future trial to overcome barriers and challenges that may be encountered in this pilot.

There is currently limited evidence supporting the choice of a primary health outcome, but the use of depressive symptoms is supported by one systematic review.[24] This pilot study will help to determine whether measuring depressive symptoms is the most appropriate primary outcome for a future trial.

One of the main challenges of this study will be to ensure that the intervention and the recruitment, training and supervision of volunteers are carried out according to the trial protocol as these aspects vary between befriending services, and was an issue in a recent trial of befriending in people with psychosis.[25] While both befriending services have committed to the study, there are resource implications for these services as the intervention costs are not covered by the research funding. External factors such as funding cuts to these services could influence the delivery of the intervention and the success of the trial. The process data will allow us to examine whether a future trial would need to be more pragmatic and whether the befriending intervention should permit more flexibility.

**Author affiliations**
[1]Division of Psychiatry, University College London, London, UK
[2]Research and Development, North East London NHS Foundation Trust Goodmayes Hospital, Ilford, UK
[3]Unit of Social and Community Psychiatry, Barts and the London School of Medicine and Dentistry, University of London, London, UK
[4]Department of Statistical Science, University College London, London, UK
[5]Primary Care and Population Health, University College London, London, UK
[6]Hackney Volunteer and Befriending Scheme, Outward, London, UK
[7]The Befriending Scheme, Sudbury, Suffolk, UK

**Contributors** AA is the chief investigator and was responsible for the overall study design and conduct. AA, SP, MK, AH, BL-E and RO contributed to the design of the study and methodology. SF, VF, DOC, CR and SM were responsible for designing and delivering the intervention. EM carried out the data collection. RO, RJ and MP contributed to the analysis plan. All the authors reviewed and approved the manuscript.

**Funding** The study/project is funded by the National Institute of Health Research (NIHR) (Public Health Research Funding Stream (project reference: 16/122/57)). The views expressed are the views of the authors and are not necessarily those of the NIHR of the Department of Health and Social Care.

**Competing interests** None declared.

**Patient and public involvement** Patients and/or the public were involved in the design, conduct, reporting or dissemination plans of this research. Refer to the Methods section for further details.

**Patient consent for publication** Not required.

**Provenance and peer review** Not commissioned; externally peer reviewed.

**ORCID iD**
Afia Ali http://orcid.org/0000-0002-0104-9370

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
