## [Reviewer comments · BMJ Open]

ARTICLE DETAILS

TITLE (PROVISIONAL)	One to one befriending for people with intellectual disability and symptoms of depression: A protocol for a pilot randomised controlled trial.
AUTHORS	Ali , Afia; McKenzie, Emma; HASSIOTIS, ANGELA; Priebe, Stefan; Lloyd-Evans, Brynmor; Omar, Rumana; Jones, Rebecca; Panca, Monica; Fernandez, Vincent; Finning, Sally; Moore, Shirley; O'Connor, Danielle; Roe, Christine; King, Michael

VERSION 1 – REVIEW

REVIEWER	Silvana Mengoni University of Hertfordshire, UK
REVIEW RETURNED	27-Sep-2019

GENERAL COMMENTS	Thank you for submitting this protocol for publication. This is an exciting study and I was pleased to review the protocol. The study is well-designed to address the research questions, and is of potential significance to people involved in delivery and commissioning of NHS and community-based services. I have a few comments regarding clarification that the authors may wish to consider:  1. P5, line 13 – Reference 24 is described as looking at studies where the befriending was delivered by ‘volunteers only’. What is this in comparison to, i.e. how does it differ to reference 23? 2. P5, line 31 – a brief description of active mentoring and how it differs from befriending would be beneficial. 3. The two follow-up timepoints are described in different ways in the manuscript, sometimes as 6 and 12-months post randomisation and sometimes as post-intervention and 6-months follow-up. Consistent terminology for the follow-up timepoints may aid the clarity of the protocol. 4. P6 sample size section – This section refers to a sample size of 50 and a sample size of 40. I think the 40 may be referring to a potential dropout of 20% but I’m not sure - please could you clarify? Lines 45-46 state that each of the two sites will recruit 4 participants per month over 6 months. This would result in 48 participants rather than 50. 5. P7, line 34 – there is a typo as this should read ‘referral form’ 6. P8, line 32 – this states that the intervention will last for one hour per week. This may be insufficient time to engage in local activities and amenities (line 34/35). Is there flexibility in the duration of the sessions if both the person with ID and volunteer agree? 7. P9, line 3-5 – will the volunteering coordinator be meeting with the volunteer and/or person with ID? If both, will this be together or separately?
--

	8. P11, line 3-4 – this states that outcome assessments will be carried out with the volunteers prior to matching and after 6 months, yet Table 2 shows three assessment timepoints. 9. P11, line 26 – please add the relevant reference 10. P11, line 42-43 – the text in parentheses doesn't make sense to me, should this read '8 per site included' rather than '8 per site including'? 11. P11, line 44 – 'at each site' can be deleted after '5-8 carers' as it is clear from earlier in the sentence that this is occurring per site.
--	--

REVIEWER	Judy A Lowthian Bolton Clarke Research Institute, Bolton Clarke; School of Public Health & Preventive Medicine, Monash University; Faculty of Health & Behavioural Sciences, University of Queensland.
REVIEW RETURNED	05-Oct-2019

GENERAL COMMENTS	Thank you for asking me to review this study protocol for a pilot RCT designed to assess acceptability and feasibility of conducting a full RCT of volunteer befriending for people with mild/moderate intellectual disability (ID). The rationale, aim and objectives are well articulated; and the methods, measurement tools and proposed analysis and evaluation are appropriate. My queries are as follows: p7 line 24 Please clarify whether the proposed befriending services/volunteer coordinators have experience with supporting people with mild/moderate ID. If not, please clarify what training and support will be provided throughout the study. P7 line 35 Please clarify whether the RA has experience with interacting with people with mild/moderate ID. If not, please clarify what training and support will be provided throughout the study. P8 line32 The intervention encompasses one hour/week for 6 months – is a one hour session feasible if the pair are to access community activities? p7 line 47 & p9 line17-30 Please clarify whether the recruited volunteers have experience with supporting people with mild/moderate ID. The training is described as face-to-face OR e-learning: is this sufficient for volunteers with no experience with people with ID; and please outline risk mitigation strategies for managing potential red flags should the recipient's mental health deteriorate during the intervention (e.g.) referral pathways to professional mental health workers etc p12 line 54 - I note that advisory groups will provide advice about the study processes, but have people with mild/moderate ID and potential volunteers been involved with designing the study?
---

VERSION 1 – AUTHOR RESPONSE

Response to reviewers' comments

Thank you for giving us the opportunity to revise the manuscript. The reviewers' comments have been very helpful in shaping the manuscript. Our response to each of the comments are summarised below.

Reviewer: 1

Thank you for submitting this protocol for publication. This is an exciting study and I was pleased to review the protocol.

The study is well-designed to address the research questions, and is of potential significance to people involved in delivery and commissioning of NHS and community-based services. I have a few comments regarding clarification that the authors may wish to consider:

1. P5, line 13 – Reference 24 is described as looking at studies where the befriending was delivered by ‘volunteers only’. What is this in comparison to, i.e. how does it differ to reference 23?

- *I have now clarified that reference 23 relates to a systematic review whereby all types of befrienders were included, including those from a health and social care background, whereas reference 24 relates to a systematic review comprising studies of unpaid volunteers only.*

2. P5, line 31 – a brief description of active mentoring and how it differs from befriending would be beneficial.

- *A description of the active mentoring intervention has been provided. In the paragraph describing the conceptualisation of mentoring, there is a brief description of the differences between befriending and mentoring.*

3. The two follow-up timepoints are described in different ways in the manuscript, sometimes as 6 and 12-months post randomisation and sometimes as post-intervention and 6-months follow-up. Consistent terminology for the follow-up timepoints may aid the clarity of the protocol.

- *Thank you for highlighting the inconsistency in terminology. I have now corrected this in the manuscript.*

4. P6 sample size section – This section refers to a sample size of 50 and a sample size of 40. I think the 40 may be referring to a potential dropout of 20% but I'm not sure - please could you clarify? Lines 45-46 state that each of the two sites will recruit 4 participants per month over 6 months. This would result in 48 participants rather than 50.

- *We apologise for the lack of clarity. The sample size of 50 refers to the number of potentially eligible participants who will be approached to take part in the study, assuming that 80% agree to take part (a recruitment rate of 80%), this will give us 40 recruited participants. I have tried to clarify this in the text. I have adjusted the recruitment figures for each site to 3.3 people per month over 6 months.*

5. P7, line 34 – there is a typo as this should read ‘referral form’

- Thank you -this typo has been corrected.

6. P8, line 32 – this states that the intervention will last for one hour per week. This may be insufficient time to engage in local activities and amenities (line 34/35). Is there flexibility in the duration of the sessions if both the person with ID and volunteer agree?

- We agree with the reviewer that there should be sufficient time for engagement. The expectation is that the volunteer and individual with ID will spend at least one hour together and a longer duration is acceptable if both parties agree. The wording has been changed to “at least one hour” to reflect this.

7. P9, line 3-5 – will the volunteering coordinator be meeting with the volunteer and/or person with ID? If both, will this be together or separately?

- This point has been clarified in the text. The volunteer coordinator will meet with both individuals together six weeks into the intervention and again before the end of the intervention. However, there will be monthly contact with each individual via telephone or face to face contact to ensure that concerns or queries can be addressed in a timely manner.

8. P11, line 3-4 – this states that outcome assessments will be carried out with the volunteers prior to matching and after 6 months, yet Table 2 shows three assessment timepoints.

- I have clarified that there will be three assessments: at baseline (prior to matching), post intervention (6 months after baseline assessment) and at six months follow-up.

9. P11, line 26 – please add the relevant reference

- Thank you for highlighting this omission. The reference (42) has now been added.

10. P11, line 42-43 – the text in parentheses doesn’t make sense to me, should this read ‘8 per site included’ rather than ‘8 per site including’?

- Thank you – I have corrected the type. It should read “8 per site”.

11. P11, line 44 – ‘at each site’ can be deleted after ‘5-8 carers’ as it is clear from earlier in the sentence that this is occurring per site.

- Thank you, I have corrected this.

Reviewer: 2

Thank you for asking me to review this study protocol for a pilot RCT designed to assess acceptability and feasibility of conducting a full RCT of volunteer befriending for people with mild/moderate intellectual disability (ID).

The rationale, aim and objectives are well articulated; and the methods, measurement tools and proposed analysis and evaluation are appropriate.

My queries are as follows:

1. p7 line 24. Please clarify whether the proposed befriending services/volunteer coordinators have experience with supporting people with mild/moderate ID. If not, please clarify what training and support will be provided throughout the study.

- I have now clarified this in the text (under “recruitment” and “befriending intervention: iv. training and support for befriending services”). Both of the participating befriending services have experience working with people with mild and moderate ID and currently arrange befriending relationships for this group. This includes advertising for volunteers, providing training to volunteers,

matching individuals with ID to volunteers and monitoring relationships. They have systems in place for highlighting and reporting concerns such as safeguarding concerns. All the members of staff participating in the research will complete Good Clinical Practice (GCP) in research training and training on study processes and procedures. Staff will have regular contact with the research team to discuss concerns and attend the Trial Management group.

2. P7 line 35. Please clarify whether the RA has experience with interacting with people with mild/moderate ID. If not, please clarify what training and support will be provided throughout the study.

-I have clarified in the text (under "Ethics, governance and dissemination) that the RA will have experience working with people with ID. Training on assessing capacity and obtaining consent, and how to carry out assessments (including IQ testing using the WASI-II) will be provided. The RA will receive regular supervision from the CI.

3. P8 line32 The intervention encompasses one hour/week for 6 months – is a one hour session feasible if the pair are to access community activities?

- I have changed the wording slightly to indicate that the pair will need to meet "at least 1 hour per week" as it would be fine for the pair to meet for a longer duration to access activities. As volunteers are giving up their time, we were mindful that a longer duration may not be practical for volunteers who may also be working full time.

4. p7 line 47 & p9 line17-30. Please clarify whether the recruited volunteers have experience with supporting people with mild/moderate ID. The training is described as face-to-face OR e-learning: is this sufficient for volunteers with no experience with people with ID; and please outline risk mitigation strategies for managing potential red flags should the recipient's mental health deteriorate during the intervention (e.g.) referral pathways to professional mental health workers etc. p12 line 54 -

-The volunteers may or may not have prior experience of working with people with ID. For pragmatic reasons, the volunteers will receive the usual training that is provided by the befriending services to their volunteers, which is a mix of face to face and e-learning. This includes training on learning disability awareness, managing the relationship and safeguarding. The training is also supplemented with a study manual and e-learning developed by the research team that provides additional information on communication skills, challenging behaviour and mental health issues and how to respond in a crisis. The volunteers will also be able to contact the volunteer coordinator if they have any concerns and will receive monthly supervision where the volunteer will have opportunities to develop their learning further. In the text I have clarified that the training will comprise both face to face and e-learning and added additional information on the topics covered.

If there are concerns about the participants' mental health, the volunteer will be advised to contact the volunteer coordinator at the relevant scheme, who will then notify the study CI and the clinician/ mental health worker who referred the participant to the study or is known to be working with the participant. In an emergency situation, out of hours, the volunteer will contact the participant's carer to inform them of the situation, and if appropriate, contact emergency services. This pathway has been included in the text under "adverse events".

5. I note that advisory groups will provide advice about the study processes, but have people with mild/moderate ID and potential volunteers been involved with designing the study?

-I can confirm that people with ID, volunteers and befriending schemes were consulted about the design of study and made suggestions that were incorporated into the final design. I have added a sentence describing this.

VERSION 2 – REVIEW

REVIEWER	Silvana Mengoni University of Hertfordshire
REVIEW RETURNED	22-Nov-2019

GENERAL COMMENTS	Thank you for addressing my comments and revising the manuscript. I have two minor requests for clarification 1. The abstract states that the target sample size is 50, but from the revised manuscript, I understood that the team aim to recruit 40 participants 2. P9 line 45 – Should ‘Good Medical Practice’ be ‘Good Clinical Practice’?
--

REVIEWER	Judy Lowthian (i) Bolton Clarke Research Institute, Bolton Clarke, Australia (ii) School of Public Health & Preventive Medicine, Monash University, Australia
REVIEW RETURNED	02-Jan-2020

GENERAL COMMENTS	Thankyou for addressing my queries. I wish you the very best with conducting this study.
---

VERSION 2 – AUTHOR RESPONSE

I would like to thank both reviewers for their time in reviewing the manuscript, and reviewer 1 for identifying the errors in the manuscript, which I have now addressed:

1. The abstract states that the target sample size is 50, but from the revised manuscript, I understood that the team aim to recruit 40 participants

- I have corrected this in the abstract, which now reads "the trial aims to recruit 40 participants"

2. P9 line 45 – Should ‘Good Medical Practice’ be ‘Good Clinical Practice’?

- The reviewer is correct. I have amended this to "Good Clinical Practice".